# 'Relieved to be seen'—patient and carer experiences of psychosocial assessment in the emergency department following self-harm: qualitative analysis of 102 free-text survey responses

Leah M Quinlivan  ,[1,2] Louise Gorman,[1,2] Donna L Littlewood,[2] Elizabeth Monaghan,[2] Steven J Barlow,[2] Stephen M Campbell,[2] Roger T Webb,[1,2] Navneet Kapur[1,2,3]

¹Centre for Mental Health and Safety, University of Manchester, Manchester, UK
²NIHR Greater Manchester Patient Safety Translational Research Centre, The University of Manchester, Manchester Academic Health Science Centre, Manchester, UK
³Greater Manchester Mental Health NHS Foundation Trust, Manchester, UK

**Correspondence to**
Dr Leah M Quinlivan;
leah.quinlivan@manchester.ac.uk

## ABSTRACT

**Objectives** We sought to explore patient and carer experiences of psychosocial assessments following presentations to hospital after self-harm.

**Design** Thematic analysis of free-text responses to an open-ended online survey.

**Setting** Between March and November 2019, we recruited 88 patients (82% women) and 14 carers aged ≥18 years from 16 English mental health trusts, community organisations, and via social media.

**Results** Psychosocial assessments were experienced as helpful on some occasions but harmful on others. Participants felt better, less suicidal and less likely to repeat self-harm after good-quality compassionate and supportive assessments. However, negative experiences during the assessment pathway were common and, in some cases, contributed to greater distress, less engagement and further self-harm. Participants reported receiving negative and stigmatising comments about their injuries. Others reported that they were refused medical care or an anaesthetic. Stigmatising attitudes among some mental health staff centred on preconceived ideas over self-harm as a 'behavioural issue', inappropriate use of services and psychiatric diagnosis.

**Conclusion** Our findings highlight important patient experiences that can inform service provision and they demonstrate the value of involving patients/carers throughout the research process. Psychosocial assessments can be beneficial when empathetic and collaborative but less helpful when overly standardised, lacking in compassion and waiting times are unduly long. Patient views are essential to inform practice, particularly given the rapidly changing service context during and after the COVID-19 emergency.

## INTRODUCTION

Self-harm is a common antecedent and strong risk factor for suicide.[1–3] Repeat self-harm occurs frequently, and people who harm themselves more than once have an even higher risk of suicide.[3] Although

### Strengths and limitations of this study

► Understanding what works and does not work for patients when receiving psychosocial assessments following self-harm is key to improving practice; however, such evidence is limited.

► This is the largest qualitative study on psychosocial assessments following self-harm and the only study to have also included carer perspectives.

► Our extensive patient and carer involvement and use of a qualitative survey enabled us to access a marginalised and stigmatised group of patients and carers with substantive unmet healthcare needs.

► A limitation of this study is the use of a non-probability survey design. However, our aim was to provide qualitative experiential data and not to generalise to the broader population.

► Most of the respondents in the study were white British Women from England (72/88, 81.8%), and their experiences may differ in important ways from other patients who have limited literacy skills or did not complete the survey.

hospital presentations represent the 'tip of the iceberg' for self-harm,[4] they provide an important opportunity for intervention via the provision of good-quality care.[5] Liaison psychiatry services are an integral part of the self-harm care pathway.[2] Specialist teams are typically situated in acute hospitals and provide liaison care for patients on wards and in the emergency department.[2] Psychosocial assessments are a core component of care and are recommended for all patients presenting to hospital services having harmed themselves.[2] Good-quality assessments may help to prevent repeat self-harm.[6 7]

Liaison psychiatry has rapidly transformed to manage the consequences of the COVID-19 pandemic on service provision.[8]

Mental health services are in flux and long-term changes are likely.[8] However, psychosocial assessments remain a core part of practice.[8] To ensure high-quality care in liaison psychiatry, an enhanced understanding from the perspective of patients and carers as to what helps and does not help is now needed more than ever.[9]

Several studies of variable methodological quality and reporting standard have investigated healthcare service experiences following self-harm.[10] However, only two relatively small and geographically local studies have focused on psychosocial assessments,[11 12] and none have included carer perspectives or involved patients and carers throughout the research process.

Between March and November 2019, we collaborated with mental health trusts in England and other community organisations to conduct a qualitative online patient and carer survey on psychosocial assessments in relation to self-harm. For this study, the specific objective was to explore positive and negative patient experiences of psychosocial assessments in the emergency department in a large sample of patients and carers.

## METHODS
### Design and sample
Taking a pragmatist research approach,[13] we conducted a qualitative online survey[14] to investigate patient and carer experiences of assessments following self-harm. Pragmatism is a solution-focused approach that prioritises the most appropriate methods to address research problems .[15] Our aim for this research was not to generalise to a wider population or to estimate prevalence, but to instead provide qualitative experiential data about the assessment process. Recruitment is often challenging for stigmatised groups of people and particularly so in relation to hospital presentations following self-harm.[10] An online survey, which was co-designed with a diverse panel of patients and carers, enabled us to access a marginalised group of people who are often stigmatised, thereby providing greater anonymity and control to participants when sharing their experiences.[14 16]

### Recruitment
We invited patients and carers aged ≥18 years with experience of self-harm (defined as intentional self-poisoning or self-injury irrespective of the suicidal intent),[2] psychosocial assessments and psychological therapies to participate in a national online survey. We did not include people under the age of 18 years because of differences in service provision for child and adolescent mental health services in England.[17] To capture a wide range of experiences, we recruited participants through 16 National Health Service (NHS) mental health trusts around England, social media (eg, Twitter, Facebook), community organisations (eg, charities, patient groups) and newsletters, from April 2019 through November 2019. We closed recruitment when there was a sufficient detailed, rich and range of responses for the free-text questions

(determined by consensus between the research team and patient/carer advisory panel) and due to pragmatic constraints (eg, study deadlines). Additional methodological information is presented in online supplemental appendix 1.

### Analysis
We used thematic analysis to explore patterns across the data set that represented participants' experiences.[14] Structured questions were used deductively to form the initial coding framework. Inductive methods were developed to capture additional codes and context across the data set.[14 18] Our patient and carer advisory group coded the data and developed the coding framework, which was applied across the data set. After immersion and familiarisation with the data, LG and LQ independently coded the full study data set with this framework and reviewed convergence for codes and themes. Throughout the iterative coding process, the researchers reviewed and revised the newly developed inductive codes and the application of the coding framework. Working together with patients and carers, themes were constructed and revised from group discussion to enhance their clarity and relevance. We analysed responses from subgroups (eg, patients/carers, gender, age groups) together because the responses consistently overlapped. The final thematic structure and illustrative quotes were agreed through discussion among the team (LQ, LG, EM, DL, SB, RW, NK).

Descriptive quantitative analyses were performed using SPSS V.22[19]; NVivo V.12 software[20] was used for data management.

### Patient and public involvement
Our patient and carer advisory panel were involved in all aspects of the research process, including the design, conduct, reporting and dissemination plans for the research. Two panel members (EM, SJB) with lived experience in this area contributed in depth to the analyses and are co-authors of this article. This research was also reviewed by a team with experience of mental health problems and their carers who have been specially trained to advise on research proposals and documentation through the Feasibility and Acceptability Support Team for Researchers: a free, confidential service in England provided by the National Institute for Health Research (NIHR)-funded Maudsley Biomedical Research Centre via King's College London and South London and Maudsley NHS Foundation Trust. There was patient and public involvement input into our dissemination plan, which includes communicating key findings to relevant patient groups, carers and mental health services.

## RESULTS
### Descriptive quantitative results
In total, 102 participants provided text responses on experiences of psychosocial assessments in the online survey.

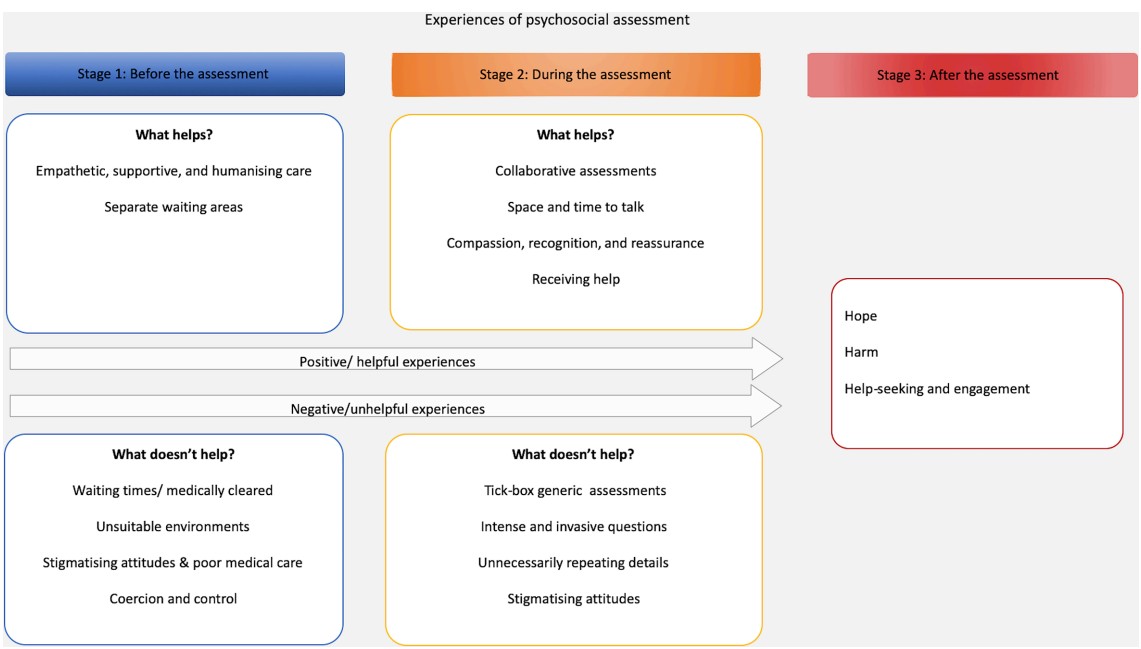

**Figure 1** Themes and subthemes across the three assessment process stages (1: before the assessment; 2: during the assessment; 3: after the assessment).

The majority of participants were patients (88/102, 86.3%), and the remainder were carers (14/102, 13.7%). Patients were aged between 18 and 75 years, and their median age was 34 years. Carers were aged between 41 and 73 years, and their median age was 56 years. Most patient (72/88, 81.8%) and carer (13/14, 92.9%) respondents were women.

## Qualitative results
### Experiences of psychosocial assessments
Psychosocial assessments are embedded in pathways of care when patients attend the emergency department after harming themselves. Participant experiences reflected the temporal and dynamic nature of presentations for self-harm (eg, the assessment process may take place over several hours). To understand what helps and does not help for assessments, it was important that we considered the context of what went before and came after the assessment. To capture the context of psychosocial assessments, the themes are presented under stages of the care: (1) before the assessment, (2) during the assessment and (3) after the assessment. Figure 1 presents the themes and subthemes throughout the process. Additional supporting quotations are in tables 1–3.

## Themes
### Stage 1. Before the psychosocial assessment: what helped?
*Empathetic, supportive and humanising care*
Positive experiences of the preassessment process centred on empathetic, supportive emergency department staff who reassured participants and humanised the experience. Many people were anxious and uncertain about the waiting times and assessment process.

Clear communication, check-ins from staff, and empathy helped to reassure and encourage people to stay for the assessment. Having support during the initial stages helped some participants feel 'looked after by a member of staff' (R94, female, age 45–49 years, patient), which was helpful when they were distressed. Soothing aids such as heat packs and/or drinks and emotional support helped to humanise the process while participants waited for the assessment:

> Other times I've been given a blanket and a hot drink, [the clinician] explained what will happen, and someone stays with me until the assessment happens. (R04, female, age 40–44 years, patient)

### What did not help?
*Waiting times/medically cleared*
Most participants reported long and frustrating waiting times for their psychosocial assessment. There was little communication about time frames, which left people unsure of what was happening, when they would be seen, by whom, etc. Often it was not possible to speak to a mental health professional until the physical injury had been treated (medically cleared), which added further delays, distress and frustration. Some participants were left alone for long periods of time while they waited to see a mental health clinician, leading to greater uncertainty about the process:

> It was not possible to speak to someone until I was 'medically cleared'. This left a lot of time alone, without support even to ask the staff for a toothbrush or understand how long I'd be there. (R70, female, age 25–29 years, patient)

**Table 1** Before the psychosocial assessment: themes and exemplar quotations

| Psychosocial assessment stage | Themes | Example quotes |
|---|---|---|
| What helped? | Empathetic, supportive and humanising care | 'I arrived at the emergency department at 2am after a breakdown and had to wait over 5 hours for the mental health team to see me on the day shift. I had self-harmed and told the staff this, and I told them that I was feeling suicidal. I didn't refer myself to the hospital and I wanted to leave during the 5-hour wait but I was encouraged to wait to see the doctor before I could leave. The nurses on shift were pleasant and one- on-one with me a few times, offered drinks, etc' (R86, female, age 19–24 years, patient). 'In the hospital they were amazing!! Very kind and helpful, looked after me extremely well and didn't make me feel like a burden at all' (R63, female, age 25–29 years, patient). |
| What did not help? | Waiting times/ medically cleared | 'If we had gone into A&E for self-harm, psychiatric liaison would refuse to see my daughter until she has completed her treatment, be it stitches, stapling or NAC. They would refuse to do an assessment until she was medically fit/ cleared; we often had to wait 5, 6, 7 hours for someone to do a mental health assessment'(R20, female, age 55–59 years, carer). |
| | Emergency department environments | 'Had to share room with a man who was very intoxicated and who had soiled himself, which was very frightening as a 21-year-old female. Then was moved to corridor for some time, until being moved back to the waiting room once my obs were stable as no more space on the ward. I was not able to see the mental health team until 10am the following day and later found out they had gone home for the night' (R41, female, age 25–29 years, patient). |
| | Stigmatising attitudes and poor medical care | 'In many cases, staff lacked compassion. Such as invalidating my distress, stigmatising responses such as 'wow you really meant to kill yourself, didn't you!!', exclaiming at the severity of my previous scarring and saying I was 'adding to the collection', saying that my pain threshold must be high and deciding not to give me any pain relief or medications when stitching or cleaning wounds (almost as if it was to be a punishment for self-harming), saying that I was 'wasting time' and other people had 'real' injuries (R17, female, age 25–29 years, patient). |
| | Coercion and control | 'Don't like it when male security escort me to toilet in case I abscond'(R01, female, age 60–65 years, patient). |

*Emergency department environments*

Some people were offered a separate waiting room and support while they waited for the assessment, which was helpful. However, for others, the emergency department environment felt physically and psychologically unsafe. The noise, intensity and lack of privacy increased their distress, particularly where there was a lack of communication over the process and waiting times. Others waited in physically unsafe rooms for long periods of time. One patient reported harming herself with 'sharp objects' left in a bin while she waited over 5 hours to see mental health practitioners (R01). One mother reported that her son harmed himself and required stitches in an unsafe side room without scrutiny or supervision' (R100, female, age 70–74 years, carer):

You are placed in a very small, usually filthy room and are left for hours. Myself and my family waited in the room 8 hours once whilst they had to try and calm me down and prevent me from absconding. (R112, female, age 35–39 years, patient)

*Stigmatising attitudes and poor medical care*

Participants reported a lack of parity of esteem in the emergency department between self-harm and physical healthcare. Negative interactions with general medical staff while waiting for an assessment or during medical treatment for self-harm were reported as common. Some felt they had been treated with contempt and as a lower priority than those with physical health issues, especially during busy periods. Others reported receiving punitive medical care from some general medical staff, which included insensitive and derogatory comments about their injuries. Some participants reported that they were refused pain relief because they had harmed themselves: 'You never had an anaesthetic when you cut, so we won't give one now to suture your wounds!' (R08, female, age 70–74 years, patient). Others were refused medical treatment because it was felt that they would repeat self-harm: 'I was told that it would be a waste of time because "you'll just go and do it again, or cut through

**Table 2** During the psychosocial assessment: themes and exemplar quotations

| | Themes | Example quotes |
|---|---|---|
| During the assessment: What helped? | Collaborative assessments and engaged communication | 'In any situation, what works well is when I feel listened to and like I had some input and agreement into the decision and follow up and most importantly that I understood the situation and why it was happening' (R34, female, age 30–34 years, patient). |
| | Space and time to talk | 'I was given a very quick psychiatric assessment in A&E. I was appreciative of being given some attention at the time as it was the first time I'd spoken about my mental health and self-harm/suicidal ideation. … Ideally it would be beneficial to be given some time/space to explore issues rather than feeling that they want you processed and out of the department as soon as possible'(R04, male, age 40–44 years, patient). |
| | Recognition and reassurance | 'The last two occasions I have had an assessment with a psychiatric-liaison practitioner, they have been really positive. I was made to feel as a human and felt as though how I was feeling was validated… Initially, I was nervous. I find talking openly very difficult and tend to fabricate answers due to being scared of the repercussions. The nurses I saw could sense my apprehensions but were encouraging with me to speak. Afterwards, I felt like I wanted them to be a constant part of my mental health recovery, knowing that the reason why I saw them was due to nearly losing my life. They temporarily restored my faith in the MH system' (R59, female, age 20–24 years, patient). |
| | Help | 'I got taken into a separate room to discuss my situation and options available - the staff member listened well and took all my intentions seriously. I felt heard. I was sent home afterwards with a detailed "crisis plan of action" that I could read and share whenever things were getting difficult. This was helpful' (R113, female, age 35–39 years, patient). |
| During the assessment: What did not help? | Generic tick-box assessments | 'What didn't work well was being told I would be okay, the nature of a checklist-like set of questions to evaluate someone's mental health, left no room for me to really talk about how I was actually feeling'(R09, non-binary, age 18–24 years, patient). |
| | Intense and invasive questions | 'These assessments were often quite intense and invasive, in so much as a professional with whom I would never again have contact sought to dredge up my entire life history, only to send me home feeling more unsettled than when I'd arrived …' (R101, female, age 30–34 years, patient). |
| | Unnecessarily repeating details | 'Sometimes there is little point in repeating the reasons or trying to explain why I have self-harmed. People often don't understand it as it is linked to OCD. I get frustrated when people don't understand and it then makes it difficult to work with professionals'(R95, female, age 25–29 years, patient). |
| During the assessment: What did not help? | Stigmatising attitudes during the assessment | 'What doesn't work is being told I am doing it for attention, and that they know better than me what is helpful, so they won't change the plan. The most unhelpful things are to be told that I didn't really want to kill myself because I'm not dead and that it is up to me if I kill myself' (R116, female, age 35–39 years, patient).<br><br>'When I have presented with a diagnosis of emotionally unstable personality disorder, triage was still quick, but staff have been cold and lacking in empathy and compassion. Assessments were treated, almost with boredom, and I've been discharged despite being a current risk of suicide or further self-harm'.(R47, female, age 25–29 years, patient). |

| Table 3 | After the psychosocial assessment: themes and exemplar quotations |
|---|---|
| **Themes** | |
| Hope | 'Before and during, I felt ashamed, like a fraud and saddened that I was putting my family and friends through all this, and even more so that I was now putting a drain on resources that could off been used for someone else who needed it more than I. After, I felt supported by family with me and the action plan we had all discussed. I felt there was a glimmer of hope in all the darkness' (R113, female, age 35–39 patient). |
| Harm | 'I felt that there was no room to explore what was actually going on, that I was given little opportunity to express myself, and like the whole process was a bit of a tick-box exercise, rather than a supportive conversation or a way to come up with helpful suggestions. 9 times out of 10 these assessments left me feeling worse' (R101, female, age 30–34 years, patient). |
| | 'The RAID assessment made me feel listened to and gave me a next step. But the next step was pointless. I was back home feeling like nothing changed. So, I felt in many ways worse' (R70, female, age 25–29 years, patient). |
| | 'They tell you to simply go listen to music or have a bath and then discharge you and as a result, you attempt to end your life'(R30, female, age 18–24 years, patient). |
| Help-seeking and engagement | 'I felt judged whilst being asked questions about how I was feeling, and like I was being misunderstood and like I wasn't being taken seriously. And after the assessment and after discharge from A&E the vague advice I had been given made me even less likely to return to A&E in future' (R09, non-binary, age 18–24 years, patient). |

our stitches"' (R17, female, age 25–29 years, patient), leaving some people at risk of further physical health complications or infection.

*Coercion and control*
Some participants attended the emergency department because of family or the police. Coercive practice from staff while waiting for the assessment compounded their lack of control in the process. Some were threatened with police involvement or a criminal record if they did not comply with assessments, which served to heighten feelings of disempowerment: 'I don't have a choice. They decide if I need it and need to stay or they'll call the police' (R105, female, age 30–34 years, patient). This was evident where security personnel of a different gender to the patient were present during toilet breaks, further exacerbating feelings of disempowerment. The use of coercion in the emergency department adversely affected some participants' engagement with the assessment process. One person described how they were more likely to refuse an assessment: 'because of coercion and threat of police being called if I leave or being sectioned if I don't comply' (R06, female, age 25–29 years, patient). This disempowering approach at the initial stage of the assessment contributed to an avoidance of future help-seeking for some participants.

### Stage 2. During the assessment: what helped?
*Collaborative assessments and engaged communication*
The way the assessment was conducted made a major difference to the person's experience. Active listening, good communication and engaging with the person facilitated the collaborative process during the clinical encounter. Assessments were more helpful when the person felt they were given control and choice (collaborative assessments). Participants felt heard and respected when they were included in decision-making and understood what was happening in the assessment:

> I got taken into a separate room to discuss my situation and the options available—the staff member listened well and took all my intentions seriously. I felt heard. (R113, female, age 35–39 years, patient)

*Space and time to talk*
Opportunity and time to talk about their distress were helpful for some participants. One participant reported that the assessment was the first time that he had spoken to anyone about his mental health or suicidal ideation (R04, male, age 40–44 years, patient). Other participants found having the space and time to talk helpful irrespective of the follow-up care that they received:

> I knew I wouldn't get any help but at the same time it was good to have the opportunity to talk to someone. (R36, female, age 30–34 years, patient)

*Compassion, recognition and reassurance*
Compassionate care was a key determinant of a helpful assessment. Many participants experienced internalised stigma, anxiety and a range of intense emotions during the assessment. Some people felt shame and guilt about presenting to hospital with self-harm, which was exacerbated by poor interactions with some medical staff. Others were nervous about the consequences of the assessment

and/or judgement from staff if they disclosed the intensity of their thoughts: 'Uncomfortable. It's hard to open up and/or share thoughts others might find concerning/abnormal' (R69, male, age 25–29 years, patient).

Assessments worked well when mental healthcare staff recognised the distress that led to the participant's self-harm and their anxiety over the process. Assessments were helpful for many participants when they were reassured that they did the 'right thing in coming to A&E' (R09, non-binary, age 18–24 years, patient). Kindness and consideration helped to establish a therapeutic relationship and alleviate some participants' anxiety during this sensitive time. Compassionate care and active listening from mental health staff helped to humanise the assessment experience and develop a therapeutic relationship:

> It [the psychosocial assessment] was good if staff were calm, considerate, but most of all compassionate. If they took time to listen and understand. (R20, female, age 30–34 years, patient)

### Help
Participants attended the emergency department to seek help after harming themselves. Assessments were evaluated more positively when participants received collaboratively developed safety plans, referrals to promptly accessible specialist services or provided follow-up phone calls from mental health staff. Actionable outcomes from the assessment helped to legitimise and validate participants' distress and reasons for seeking help:

> At last someone was trying to help. (R39, male, age 70–74 years, patient)

### During the assessments: what did not help?
#### Generic tick-box assessments
Unhelpful assessments were overly standardised and generic for some participants. Some participants felt that the assessments were designed to fit their psychological distress and complex experiences into 'neat little boxes' (R106, male, age 40–44 years, patient). In these instances, the focus was on ticking boxes rather than hearing what the person had to say, and little room was left to explore and assist the person with their immediate distress in this 'bureaucratic and uncaring' process (R86, female, age 18–24 years, patient). The lack of opportunity to meaningfully talk about their distress negated the potential therapeutic value of the assessment for some people:

> It was very superficial with little in-depth questioning. No real understanding that there was a real issue. (R61, female, age 50–54 years, carer)

#### Intense and invasive questions
Assessments could be intense and intrusive for some people. The combination of overly standardised assessments and a lack of therapeutic engagement made it difficult for some people to talk about their distress. In some cases, the intensity of the assessment left people feeling more vulnerable:

> I've felt emotionally uncontained due to assessments being invasive and asking you to dredge up past unhappy and traumatic experiences. I've often left feeling worse than when I went in and it put me off going to the ED when I was in a crisis for over a year. (R47, female, age 25–29 years, patient)

#### Unnecessarily repeating details
For others, assessments placed too much emphasis on the past and did not focus on their current level of distress. Many shared their life experiences and the reasons behind their self-harm during previous assessments. Being asked to give this information again exacerbated the lack of continuity of care, assumptions that notes have not been read and frustrations with the participants' experience of continuously poor understanding of some mental health conditions among some staff:

> Tedious having to explain 'whole' story each admission to a different hospital. (R28, female, age 60–64 years, carer)

> I felt like I had to go over my whole story in detail even though it could have been looked up on RIO. I found it quite intimidating. (R37, female, age 30–34 years, patient)

#### Stigmatising attitudes during the assessment
Stigmatising attitudes changed slightly across the assessment stage. Misconceptions about self-harm and attention seeking were an issue throughout the process though. However, during the preassessment stage, stigmatising attitudes often focused on the act of self-harm and a lack of parity of esteem between self-harm and physical health injuries. Participants reported receiving negative comments about wasting healthcare resources and their injuries. During the assessment stage, stigmatising attitudes commonly focused on psychiatric diagnosis and patterns of service use for many people. Many felt judged during the assessment because of their self-harm, particularly if they had previously presented to services. Some participants described issues that centred around self-harm as a behaviour and that seeking help indicated 'attention seeking behaviour' (eg, R08, female, age 75–79 years, patient; R20, female, age 55–59 years, carer). Perceiving self-harm as a behaviour indicated that the person was at fault for hurting himself or herself with little regard for any trauma or other psychological determinants. Some participants felt that they were spoken to condescendingly because they were 'misbehaving' (R95, female, age 25–29 years, patient).

Many participants with a diagnosis of personality disorder reported experiencing stigmatising attitudes from staff once they became aware of their diagnosis as if this indicated that they must be 'attention seeking' and a 'time-waster' when they sought help in the emergency

department (R20, female, age 55–59 years; carer R17, female, age 25–29 years, patient). Some participants reported that the compassion and interest in the person changed during the assessment because of their diagnosis. Many participants felt unsupported and judged because of their service use. Assessments were rushed, and their diagnosis negated any possible risk of further self-harm and suicide from the perspective of healthcare staff:

> No real understanding that there was a real issue - felt like my daughter was being treated as an attention seeker. (R61, female, age 50–54 years, carer)

### Stage 3. After the assessment: consequences
*Hope*

Feelings after the assessment were closely related to the perceived quality of the assessment and support received during and afterwards. More positive emotions were reported after collaborative and compassionate assessments that provided practical help from the mental health clinician. Humanising care helped to lessen the person's distress and ease the acuity of their psychological crisis. Emotive challenges of the assessment process were offset for some because they received a first step towards seeking help and managing their self-harm. Humanising and supportive care helped to provide participants with hope:

> After the assessments that were more caring and humanising, I tended better to my injuries, and felt less self-hatred. They were still deeply upsetting and difficult experiences, but I was more likely to be able to take steps toward self-care and getting help, and also less likely to self-harm in subsequent days. (R17, female, age 25–29 years, patient)

*Harm*

Most participants felt they would get help from the assessment, but over a third indicated that they were harmed or felt worse by the process. Some reported feeling worse after the assessment because they experienced a lack of therapeutic engagement and support. Abrupt transitions out of the assessment left other people feeling abandoned and discarded. Positive effects of the assessment were negated for some people, with some participants reporting a worsening of their distress.

Negative and frustrating experiences of aftercare and support led many participants to perceive the assessments as a 'waste of time' (R128, female, age 18–24 years, patient). Receiving out-of-date signposting information was frustrating for some people, and others found the clinician's advice to be overly simplistic. Feeling unsupported during and after assessments led to despondency, hopelessness and increased distress for some participants. In several instances, negative experiences of the assessment process and lack of support led to further thoughts and acts of self-harm:

> The assessment feels like, & is, a waste of time & makes you feel even more ashamed & worthless & more likely to self-harm again. (R21, female, age 45–49 years, patient)

### Help-seeking and engagement

Experiences of attending the emergency department following self-harm had a cumulative effect and resulted in far-reaching consequences on help-seeking and engagement with mental health services for some participants. Some reported that the lack of a therapeutic assessment, inadequate follow-up care and/or long waiting lists to access support increased their levels of distress and disillusionment. Over a third of participants reported that they were less likely to seek help or engage with assessments and/or were more likely to leave the emergency department without an assessment:

> I didn't feel like I was being treated as a very high priority, and went home feeling like there's no point in going to A&E when I'm suicidal from now on (even though that's what all health professionals seem to suggest). (R15, male, age 18–24 years, patient)

## DISCUSSION
### Main findings

We found that the psychosocial assessment process was experienced as helpful on some occasions but harmful on others. Participants reported feeling better, less suicidal and less likely to repeat self-harm after receiving compassionate care and collaborative assessments that focused on building a therapeutic relationship and providing help. Conversely, the pathway in the emergency department and the manner in which the assessment was conducted may have resulted in iatrogenic harm for some patients. Reports of substandard medical care and derogatory comments were common among participants and compounded by what appear to be preconceived ideas among some staff members regarding service use patterns, self-harm repetition and perceived attention seeking behaviour. For some participants, cumulative experiences of long-waiting times, stigmatising attitudes, disjointed care, and overly standardised assessments contributed to heightened distress, disengagement and, in some cases, further self-harm episodes.

### Strengths and limitations

We conducted an online survey with non-probability sampling to recruit our participants, an approach that is subject to potential selection bias.[21] However, our aim was not to generalise to a broader population but to provide qualitative contextual experiences of attending the emergency department for psychosocial assessments. Self-harm is often stigmatised. Recruiting individuals who have harmed themselves can be challenging, especially for emergency department studies.[10–12 17] Our co-designed qualitative online survey enabled us to investigate

a marginalised group of people to ensure that their experiences and perceptions of care are considered for service development. Over 100 people contributed their experiences of assessments and provided recommendations for improving practice. Participants were able to disclose their experiences in their own time, at their own discretion and under their own control. While the survey was designed with an emphasis on the generation of free-text qualitative data, we were unable to probe responses or explore complex issues such as the relationship between the socioeconomic status of the patient and the quality of psychosocial assessments.

We have included a demographic profile of respondents to evaluate the relevance of our findings to clinical and carer groups. Our sample included a wide age range, but only 17 (19.3%) patients were aged 18–25 years and 5 (5.7%) were aged >60 years. Experiences were similar across age groups, but future research is needed to explore issues for younger and older adults when accessing services. We did not include patients under the age of 18 years because of distinctness of service provision for this demographic group. This is an important area for further research, given the transition between adolescent and adult services.[22] Our sample is similar to Multicentre Monitoring Studies of Self-Harm in England[23] in terms of age, but we recruited more women, which is consistent with community surveys of self-harm.[10–12 24] However, self-harm and suicide rates are rising among women.[25] Our results provide important information on the unmet needs for women attending the emergency department following self-harm. Our study included relatively small numbers of carers (13.7%), black and minority ethnic groups (4%) and patients who were men (16%), which is similar to community studies of self-harm.[10 24] While experiences of assessments were consistent across subgroups in our data, help-seeking and quality of care may vary considerably for some patients in these groups compared with white British women.

Only four participants had no self-reported psychiatric diagnosis (see online supplemental appendix 1), and many had been diagnosed with a personality disorder or complex post-traumatic stress disorder. Participants had all undergone psychosocial assessment, so may be more representative of patients with a greater level of need. Further research is needed to understand the needs of patients who present to hospital emergency departments and do not have a psychiatric diagnosis or history of mental health service use.

Qualitative surveys are a useful tool for gathering data on patient experiences.[14 16] However, there is potential to further exclude under-represented populations in research, such as non-English- speaking participants, or those with literacy issues, limited digital awareness or access, and people who do not wish to write about their experiences. Language around healthcare service use and help-seeking may also seem too abstract for some marginalised individuals. Further co-designed stand-alone studies and arts-led approaches specifically designed

for these groups are indicated, to provide evidence for provision of more equitable access to services and better tailored treatments.

This is a qualitative study based on patients' experiences and their perceptions of care that they received. Perceptions of care may be affected by previous negative experiences, poor follow-up and psychological distress. Some individuals may perceive staff negatively and disengage with the assessment process if they have previously experienced stigmatising attitudes and/or have low self-worth.[12] Participants may also be more likely to participate in a study if they have negative experiences of services. However, negative experiences provide important opportunities for learning and improving services. Participants also provided examples of what they find helpful and recommendations for improving clinical practice; therefore, we can also learn from scenarios in which participants have reported positive experiences.

This is the largest qualitative study to have examined psychosocial assessment following self-harm presentations to the emergency department. Our study was strengthened by substantive patient and carer involvement throughout the research process. Analyses were conducted by a multidisciplinary team that included clinical and research expertise and people with lived experience. This enabled triangulation at a research and respondent level, which enhanced the robustness and validity of the findings.[26]

Our results provide rich and detailed information on the assessment process from a population at elevated risk of self-harm repetition and suicide.[1] We have addressed many limitations of previous studies in this area (eg, small sample sizes and lack of demographic information)[10] and the lack of patient/carer involvement. Our broad recruitment strategy included hospitals and community groups to gather a wide range of experiences from around the UK.

### Comparisons with existing research

There appears to be little change in patient experiences of emergency departments after self-harm over the last 15 years.[10–12] Our findings are consistent with largely negative accounts of patients' experiences of 'hostile care' in emergency departments following self-harm.[27–34] For example, similar to our respondents, Pembroke[28] reported the harm arising from the lack of compassionate care in the emergency department:

> The whole experience of Accident & Emergency is degrading for self-harmers. It just perpetuates the vicious circle. I am made to feel that I'm wasting their time and resources which results in me hating myself even more. This sustains my cutting. If only Accident & Emergency staff could realise that it would be easier for them and for us if they treated us humanely … (Helen, p. 23)[28]

The concerning incidents where people reported not being offered pain relief for self-harm are consistent with what was found in other studies previously and

**Table 4** NICE quality standards for self-harm compared with the study results and potential implications for practice

| | NICE quality standards (QS34) | Study results (N=102) | Potential implications for practice based on patient/carer recommendations |
|---|---|---|---|
| 1 | People who have harmed themselves are cared for with compassion and the same respect and dignity as any service user. | People experienced significant levels of stigmatising attitudes throughout the assessment process. | Given the levels of stigmatising attitudes reported by participants and elevated risk of suicide associated with self-harm and clinical populations (eg, autism spectrum condition, obsessive compulsive disorder, post-traumatic stress disorder, personality disorders),[42] staff education that also considers reflexivity, culture, and socioeconomic factors may be helpful. Training may be more effective if tailored towards staff groups (eg, acute staff, liaison psychiatry staff), co-designed and delivered by people with relevant lived experience. |
| 2 | People who have self-harmed have an initial assessment of physical health, mental state, safeguarding concerns, social circumstances and risk of self-harm repetition or suicide. | Participants felt initial assessments were focused on their mental state without attention to their distress or reasons for self-harm. Long waiting times were compounded by the continued use of having to wait until the patient is 'medically cleared' prior to assessment. | Emphatic care at initial stages may help to encourage people to stay for further assessment. Clear communication about the purpose of initial assessment and roles of each staff member may clarify the process and expected outcomes. Trained support workers or mental health volunteers could ease transitions by providing support, check-ins and soothing aids, where needed. Joint working between acute and liaison staff from initial stages could help with patient flow, engagement, and experience.[43] |
| 3 | People who have self-harmed receive a comprehensive psychosocial assessment. | Participants were unsure of the purposes of the assessment. People hoped for help but often felt let down by the lack of therapeutic engagement and aftercare. The way the assessment was carried out affected how the person felt afterwards. | Clear communication about the process, purpose and expectations for the assessment may help people to understand the process. Care and sensitivity could help people feel safer during this vulnerable stage. Collaborative assessments that focus on building a therapeutic relationship could engage people in the process and build up trust in mental health services.[44] |
| 4 | People who have self-harmed receive the monitoring that they need while in the healthcare setting, to reduce self-harm repetition risk. | Empathetic check-ins from staff or support workers helped to encourage people to stay for assessment and provide reassurance about the process. However, monitoring from security guards was experienced as perceived as coercive and affected engagement with the assessment process. | Our results suggest that the use of security guards to monitor patients may harmfully impact levels of engagement and help-seeking. Other ways to check-in and monitor patients at risk may be helpful such as trained support workers and staff. Where detainment may be necessary, staff training in mental health, robust legal justification and clear communication over the roles of the personnel involved may help to improve patient and staff experience. |
| 5 | People who have self-harmed are cared for in a safe physical environment while in the healthcare setting to reduce self-harm repetition. | Participants described poor experiences of waiting for lengthy periods of time in unsafe healthcare environments. | Environments that are separate from the emergency department may be beneficial for physical and psychological safety. Having the opportunity to wait in separate environments or quite rooms could allow some recovery and distance from the noise and intensity of the emergency department, which may facilitate greater engagement in the process. |

Continued

| | NICE quality standards (QS34) | Study results (N=102) | Potential implications for practice based on patient/carer recommendations |
|---|---|---|---|
| 6 | People receive continuing support for self-harm have a discussion with their healthcare professional about the potential benefits of psychological interventions specifically structured for people who self-harm. | Follow-up care was a major source of disillusionment for participants. There was little discussion of psychological therapies for people who have harmed themselves during the psychosocial assessments. Long waiting times to access psychological therapies were common. Participants were desperate for help at the time of the assessment and for more therapeutic engagement. | Our results suggest that greater communication and transparency over psychological therapies and waiting times may be helpful. Therapeutic assessments and enhanced availability of psychological therapies delivered by the liaison psychiatry team may be beneficial for some patients and carers. |
| 7 | People receiving continuing support for self-harm and moving between mental health services have a collaboratively developed plan describing how support will be provided during the transition. | Poor transitions while in the emergency department and linking between services (eg, primary care, secondary care) left many participants feeling distressed and abandoned. Some participants were unclear of the role of care plans that had directives for 'do-not-assess', when attending the emergency department for self-harm. | Our findings suggest that assessments should be offered for every hospital presenting self-harm episode and plans updated. Co-designed advance directives with accessible crisis may provide more control and understanding over assessment/treatment options for some patients.[44]<br><br>Given the challenges in accessing specialist services, emergency department presentations for self-harm provide an important opportunity for intervention at a time of crisis for the patient. Good-quality, compassionate assessments and collaboratively developed safety plans may help to ease acute distress and prevent repeat self-harm.[45] Training in clinician–patient communication may help to provide a shared understanding of patient issues, reduce miscommunication and thereby enhance therapeutic engagement and quality of care.[46 47] |

NICE, National Institute for Health and Care Excellence.

patient reports.[10 27 32] Future studies should focus on the relationship between such incidents and patient safety following self-harm. Stigmatising attitudes of healthcare staff towards people who have harmed themselves and consequences for patients are well documented.[35–37] Similarly, our findings highlight pervasive negative attitudes towards people who have harmed themselves and also poor communication during the assessment process by some healthcare staff. They indicate that negative interactions with staff contributed and reinforced feelings of shame and internalised stigma. Further studies should explore the relationship between healthcare policies, cultures, systematic issues, and stigmatising attitudes towards people who have harmed themselves from a patient safety perspective.

Consistent with Hunter et al,[11] we found that assessments could positively or negatively affect hopelessness and engagement with services. Patients reported uncertainty about the purpose of the assessment, and their experiences with aftercare were often negative. Similar to Owens et al,[27] our results highlight how the transition through the emergency department affects engagement, emotional states during and after the assessment, and future help-seeking. Similar to MacDonald and colleagues,[10] we found that transitions were poorly managed for some participants, leading to uncertainty, and feelings of abandonment by the end of the assessment.

### Implications for clinical practice and policy
The novel findings that have been uncovered by this large qualitative study of patient/carer experiences of psychosocial assessment in the emergency department following self-harm suggest that elements of current practice are contrary to the National Institute for Health and Care Excellence (NICE) clinical recommendations[2] and national quality standards (QS34).[38] Greater public education on self-harm, clinical reflexivity and good-quality supervision may help to address some of the stigmatising attitudes in this area and improve patient experience. Table 4 compares the study's findings and their potential implications for practice.

Some of the poor experiences of care and assessment reported in this study possibly reflect a gap between current guidelines and their implementation, but others may reflect

more fundamental challenges not just for guidelines but for clinical practice as a whole. For example, how do we balance the safety of service users with the provision of care that is as compassionate as it can be? What are the most appropriate outcome measures not just for self-harm research but also for clinical services? The NICE guidelines for the care of people who have harmed themselves are currently under revision and represent an opportunity to hopefully address some of these issues.

Measures defined by patients, other than risk of repeat self-harm and suicide, may also help to improve patient experience.[39] To ensure cultural relevance, future co-designed studies should include the views of patients from different linguistic/cultural backgrounds in policy developments.[40] Further work is necessary to investigate clinician interpretation and implementation of guidelines in practice, alongside co-designed quality improvement initiatives that more closely evaluate patient experience in the emergency department.

Self-harm itself remains a policy priority, both as part of wider suicide prevention and in its own right.[41] There needs to be a focus on research, but also putting what we know into practice– assessments and evidence-based interventions should be available and accessible. Both staff training and service development must, of course, be guided by those with lived experience.

Liaison psychiatry services have rapidly transformed to develop alternative models of care during the COVID-19 pandemic, including ward reconfigurations and diversions away from the emergency department.[8] Psychosocial assessments remain a core component of practice, but the method and location of delivery may change (eg, virtual/ telephone assessments). The Royal College of Psychiatrists' recent survey of liaison psychiatry service changes during the COVID-19 pandemic suggests that the stigmatisation of mental illness by acute hospital staff may increase during this time of change.[8] It is therefore imperative that we better understand and ameliorate the harmful impact of stigmatising attitudes, and that any service changes consider patient and carer experiences and perceptions of care.

**Acknowledgements** We wish to thank the participants who have taken the time to generously share their often-painful experiences and also our patient and carer panel (Dawn Allen, Stephen Barlow, David Daniels, Dan Stears, Elizabeth Monaghan, Paula Mazzucco, Manoj Mistry, Tracy Neil, Fiona Naylor, Javed Rehamn, Jonathan Smith) for their involvement and insights throughout the process. We are also grateful to our GM NIHR PSTRC staff stakeholder panel and to the Feasibility and Acceptability Support Team for Researchers panel for their input into the study. We also wish to thank staff at the mental health trusts and clinical research networks, as well as community organisations for facilitating the research and assisting with the recruitment. Finally, we wish to thank the reviewers for enabling us to improve the manuscript with their helpful suggestions.

**Contributors** All authors made substantial contributions to the study. SC, RW and NK were responsible for funding acquisition. We designed and developed the study through discussion at team meetings with LQ, DL, RW and NK, and at meetings with members of our patient/public involvement (PPI) group. LQ coordinated data collection with assistance from DL. LQ and LG did the analyses with input from our wider PPI panel and from EM, SB, DL, RW and NK. LQ interpreted the results with input from LG, EM, SB, DL, RW and NK, and LQ wrote the first draft. All authors

contributed to subsequent drafts and approved the final version. All authors take responsibility for the integrity of the data and accuracy of the data analysis. NK is the guarantor of the study.

**Funding** This work was funded by the National Institute for Health Research (NIHR) Greater Manchester Patient Safety Translational Research Centre (reference number: PSTRC-2016-003). The views expressed are those of the authors(s) and not necessarily those of the NIHR or the Department of Health and Social Care.

**Competing interests** NK is a member of the Department of Health's (England) National Suicide Prevention Advisory Group. He chaired the NICE guideline development group for the long-term management of self-harm and the NICE Topic Expert Group (which developed the quality standards for self-harm services). NK is currently chair of the updated NICE guideline for Depression and Topic Advisor to the new NICE self-harm guideline. He is also supported by the Greater Manchester Mental Health NHS Foundation Trust.

**Patient consent for publication** Not required.

**Ethics approval** The authors assert that all procedures contributing to this work comply with the ethical standards of the relevant national and institutional committees on human experimentation and with the Helsinki Declaration of 1975, as revised in 2008. All procedures involving human subjects/patients were approved by Greater Manchester Central Research Ethics Committee (REC No: 18/NW/0839).

**Provenance and peer review** Not commissioned; externally peer reviewed.

**Data availability statement** No data are available. The data that support the findings of this study are not publicly available due to restrictions of the research (consent and information that could compromise the privacy of some research participants).

**ORCID iD**
Leah M Quinlivan http://orcid.org/0000-0002-3944-3613

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
