## [Reviewer comments · BMJ Open]

ARTICLE DETAILS

TITLE (PROVISIONAL)	"Relieved to be seen" - patient and carer experiences of psychosocial assessment in the emergency department following self-harm: qualitative analysis of 102 free-text survey responses
AUTHORS	Quinlivan, Leah; Gorman, Louise; Littlewood, Donna; Monaghan, Elizabeth; Barlow, Steven; Campbell, Stephen; Webb, Roger; Kapur, Navneet

VERSION 1 – REVIEW

REVIEWER	Gregory Armstrong University of Melbourne, Australia
REVIEW RETURNED	05-Oct-2020

GENERAL COMMENTS	Thank you for the opportunity to review this manuscript, which reports on analyses of data from a qualitative online survey of people who have attended an emergency department after self-harm. The paper provides some really helpful insights into people's experiences of psychosocial assessments, in particular the things that were and weren't helpful before, during and after assessments. The matching of results and recommendations against NICE guidelines was particularly helpful. My remarks are below: 1. Abstract: The final sentence of the results is confusing and needs to be revisited2. There doesn't seem to be an Appendix 13. Appendix 2 contains largely methodological information that ought to be integrated into the main body of the manuscript. Perhaps the paper was originally prepared for a journal with a tight word limit. For an open access journal, it would seem a little unnecessary to have to access basic methodological information through an Appendix.4. There are two figures that are both marked as 'Figure 1'5. The authors have elected to do an online qualitative survey. However, in-depth interviews of semi-structured interviews might have provided richer data, with prompts and conversational queues, etc. As a result of using the survey approach, the data are not particularly rich as an interviewer has not been able to probe to more deeply understand people's experiences. Nonetheless, there are some useful take home messages to inform services system development and clinician training. I simply think this could receive some acknowledgement in the limitations section, alongside the discussion around the strengths of the approach that was used.6. A sentence or two on the pragmatist epistemological approach would be helpful for those who are less familiar with this.
---

	7. The authors state that recruitment was closed when a sufficiently rich range of responses were obtained. How and by whom was this determined? 8. How do the demographics of the sample compare to the average demographics of people who attend ED after self-harm in the study location? 9. It ought to be more fully acknowledged that there were only a small number of male respondents. While representativeness was not the objective, it would be hard to disagree that the study would have benefitted from using a sampling approach to obtain a more equal proportion of male respondents. One could argue that if the study weren't able to recruit males then the objective should have changed to focus exclusively on issues for female patients. This could be reflected in the title: "...survey responses from a largely female sample". 10. What was the cultural and linguistic background of participants, and how did this influence the receipt of care in the ED? Our recent paper (see below) in the Australian context, was based around the importance of culture in psychosocial assessments for Aboriginal and Torres Strait Islander people. Leckning B, Hirvonen T, Armstrong G, Carey TA, Westby M, Ringbauer A, et al. Developing best practice guidelines for the psychosocial assessment of Aboriginal and Torres Strait Islander people presenting to hospital with self-harm and suicidal thoughts. Aust N Z J Psychiatry. 2020;4867420924082. 11. Were participant characteristics associated with particular types of responses in any way? This might be hard to reflect on given the lack of diversity in the sample. 12. Were there any discussions around the use of a peer support workforce for some elements of the process? 13. The discussion section seems a little light on, and most heavily focused on study limitations rather than discussing the results. For example, one theme that emerged was people's experiences of stigmatising attitudes among clinicians. This is one of a few critically important issues that could have been given their own paragraphs in the discussion section, with relevant references brought in. 14. Shouldn't the title refer to "psychosocial assessment" rather than "mental assessment". This paper talks about people's experiences being that the assessment was almost exclusively focused on mental health, rather than being a more holistic psychosocial assessment that reflects the issues experienced by the patient. The title unfortunately brings it straight back to mental health again. 15. Were there many participants who had no prior history or diagnosis of a mental disorder?
--	---

REVIEWER	Amy Chandler University of Edinburgh, UK
REVIEW RETURNED	08-Nov-2020

GENERAL COMMENTS	This is a strong paper that sheds light on an important issue that has long been highlighted as a problem by service users: engagement with staff in A&E when attending for self-harm, and particularly relating to the experience of psychosocial assessment. It is especially encouraging and to be commended that service user/lived experience perspective is so well embedded in project design, analysis and write up. The meaningful involvement of
---

service users/those with lived experience shows – this is a refreshingly written piece that manages to avoid the tendency to objectify and ‘other’ patients who self-harm that I still see too often in clinically based research of this type.

Using a qualitative survey of this type, with open ended questions, is a suitable method of research to address the research questions. Methods of analysis are well described and suitable.

That said, a potential limitation of the online survey approach is that it privileges participants/patients who are more literate and able to express themselves through the written word. This may have inadvertently excluded important perspectives – from patients with more limited grasp of English (whether because English is not their first language, they have language processing difficulties, or they do not like writing). It may be worth further reflecting on how recruitment and study design may further entrench the likelihood of participants being relatively similar (female, white, having a mental health diagnosis).

It is good that the authors acknowledge the limitations of their sample being almost entirely composed of white British females. This is indeed a limitation, and also often seen in online studies of self-harm. I note that employment status was measured, but wondered whether this had any impact on responses at all?

An important contribution of the paper is being able to draw out nuances in how patients experienced psychosocial care. This highlights the vital role of experience, and *how* care is provided. It relates well to existing research which shows relatively enduring negative attitudes towards those who self-harm among clinical practitioners, especially towards some patient groups (e.g. Timson et al 2012).

The authors note that ‘We are unable to say if substandard or iatrogenically harmful care caused some participants to repeat self-harm’. However, I wanted to push this issue: is it not enough to affirm that receiving care experienced as harmful is harmful, without further ‘proving’ this via actual further self-harm?

Given this issue has been addressed for such a long time, I wonder if a bit more of a historical context could be acknowledged, perhaps through reference to earlier, pioneering user led work on the challenges and harms resulting from poor medical care for people who self-harm e.g. Louise Pembroke, and perhaps also Sam Warner, and Helen Spandler’s earlier work here, as well as a study by Hadfield and colleagues that addressed some of the challenges in terms of ‘staff attitudes’ preventing provision of compassionate care for those who self-harm (Hadfield et al., 2009; Pembroke, 1998; Spandler & Warner, 2007; Warner & Spandler, 2011)?

In conclusion the authors call for more resources and training I wonder if this call could be framed in slightly stronger terms, and/or further interrogated: we have had this call for over 30 years... (the first Department of Health guidance for self-harm that I am aware of was published in 1981!). I hope that an article highlighting (again) the harms caused to patients who self-harm when they are treated badly in A&E settings will help, but I must admit to some scepticism that any of this will make a great deal of

difference. Earlier reviews by Saunders and colleagues (K. E. Saunders & Smith, 2016; K. E. A. Saunders et al., 2012) found some improvement in staff attitudes following training - but notably patient perspectives were rarely sought. This is a key area that this paper contributes to, along with the focus on experiences of psychosocial assessment specifically. However, I would suggest there is a need to go beyond staff training, and to consider more significant changes, reflecting on how policies, structures and systems, as well as speaking more broadly to societal attitudes towards mental illness might need to shift in order to improve patient experience/care? What about more robust monitoring of patient experience of A&E when attending for mental health/self-harm, as well as active engagement of patients in this process?

The section where the authors map their recommendations to the NICE guidelines is very helpful, and serves to further reinforce the challenging nature of the study findings, and I imagine may be useful for practitioners or service managers. However, again for me this served to highlight how far there is to go in terms of enacting these guidelines. This also highlights perhaps some challenging aspects of the guidelines themselves, which I acknowledge may be beyond the scope of this particular study. For instance, one of the guidelines addressed is "People who have self-harmed receive the monitoring that they need whilst in the healthcare setting, to reduce self-harm repetition risk." – this focuses attention of healthcare practitioners around surveillance with a view to 'reduction of repetition risk' – which may not sit so neatly with the provision of compassionate care for a patient who might be expected to repeat. Perhaps different measures of treatment success are needed in order to better support and reward practitioners to provide compassionate, non-judgemental care?

Overall, this paper makes a strong contribution, and highlights (again) the negative experiences of some patients who self-harm, as well as highlighting how less judgemental and compassionate care can be extremely beneficial. That said, I would suggest the recommendations could be further strengthened - could the authors consider some recommendations for policy as well as practice? Future work should also certainly look to move beyond this relatively narrow sample, to engage a more diverse patient group.

References

- Hadfield, J., Brown, D., Pembroke, L., & Hayward, M. (2009). Analysis of Accident and Emergency Doctors' Responses to Treating People Who Self-Harm. *Qualitative Health Research*, 19, 755-765.
- Pembroke, L. (1998). Only scratching the surface. *Nursing Times*, 94, 38-39.
- Saunders, K.E., & Smith, K.A. (2016). Interventions to prevent self-harm: what does the evidence say? *Evidence Based Mental Health*.
- Saunders, K.E.A., Hawton, K., Fortune, S., & Farrell, S. (2012). Attitudes and knowledge of clinical staff regarding people who self-harm: A systematic review. *Journal of Affective Disorders*, 139, 205-216.
- Spandler, H., & Warner, S. (Eds.) (2007). *Beyond Fear and Control: working with young people who self-harm*. Ross-on-Wye: PCCS Books.

	Timson, D., Priest, H., & Clark-Carter, D. (2012). Adolescents who self-harm: Professional staff knowledge, attitudes and training needs. Journal of Adolescence, 35, 1307-1314. Warner, S., & Spandler, H. (2011). New Strategies for Practice-Based Evidence: A Focus on Self-Harm. Qualitative Research in Psychology, 9, 13-26.
--	---

VERSION 1 – AUTHOR RESPONSE

Reviewer: 1

Reviewer Name

Gregory Armstrong

Institution and Country

University of Melbourne, Australia

Please state any competing interests or state 'None declared':

None declared

Comments to the Author

Thank you for the opportunity to review this manuscript, which reports on analyses of data from a qualitative online survey of people who have attended an emergency department after self-harm. The paper provides some really helpful insights into people's experiences of psychosocial assessments, in particular the things that were and weren't helpful before, during and after assessments. The matching of results and recommendations against NICE guidelines was particularly helpful.

Response:

Thank you for this positive feedback and for your helpful suggestions as to how we might improve the manuscript.

Comments:

My remarks are below:

1. Abstract: The final sentence of the results is confusing and needs to be revisited

Response:

Many thanks for pointing this out. This sentence is amended.

Reviewer comment:

2. There doesn't seem to be an Appendix 1

Response:

Thank you for highlighting this error. Appendix 1 contains the additional methodological information. Appendix 2 is now removed.

Reviewer comment:

3. Appendix 2 contains largely methodological information that ought to be integrated into the main body of the manuscript. Perhaps the paper was originally prepared for a journal with a tight word limit. For an open access journal, it would seem a little unnecessary to have to access basic methodological information through an Appendix.

Response

The word count for BMJ Open is 4000 words which is challenging when reporting qualitative research findings. The word count for the paper when submitted was 3984 words, but this has increased to accommodate extra material that has been requested the reviewers. In the text of the revised manuscript we have provided clear signposting as to where in the appendix the information can be found. To keep the manuscript with the journal's word limit boundaries, it is essential that we place some of the additional material in the appendix.

Reviewer comment

4. There are two figures that are both marked as 'Figure 1'

Response

Thanks for highlighting this error, which is now amended.

Reviewer comment

5. The authors have elected to do an online qualitative survey. However, in-depth interviews of semi-structured interviews might have provided richer data, with prompts and conversational queues, etc. As a result of using the survey approach, the data are not particularly rich as an interviewer has not been able to probe to more deeply understand people's experiences. Nonetheless, there are some useful take home messages to inform services system development and clinician training. I simply think this could receive some acknowledgement in the limitations section, alongside the discussion around the strengths of the approach that was used.

Response

Many thanks for this comment. We have now added additional quotations to illustrate the depth and richness of the data in text and in the Tables. In the Discussion, we have now highlighted that, by conducting a survey, were unable to probe and prompt participants especially in relation to nuanced issues (page 20-21 under 'Strengths and imitations').

Reviewer comment:

6. A sentence or two on the pragmatist epistemological approach would be helpful for those who are less familiar with this.

Response:

We have now provided more information on the pragmatist epistemological approach on page 6, under 'Design and Sample'.

Reviewer comment:

7. The authors state that recruitment was closed when a sufficiently rich range of responses were obtained. How and by whom was this determined?

Response:

This was determined by consensus during several time periods between the patient and carer advisory panel and research team. This is now stated in the Methods section, under recruitment on page 6-7).

Reviewer comment:

8. How do the demographics of the sample compare to the average demographics of people who attend ED after self-harm in the study location?

Response:

There is a lack of high-quality data on all emergency department presentations in the UK.¹ However, compared the Multicentre Monitoring Studies of Self-Harm in England,² our sample is similar in terms of age but we recruited more women, which is consistent with community surveys of self-harm.³ Participants had also all undergone psychosocial assessment, so many be more representative of patients with a greater level of need. We have acknowledged these points in the Discussion on page 22.

Reviewer comment:

9. It ought to be more fully acknowledged that there were only a small number of male respondents. While representativeness was not the objective, it would be hard to disagree that the study would have benefitted from using a sampling approach to obtain a more equal proportion of male respondents. One could argue that if the study weren't able to recruit males then the objective should have changed to focus exclusively on issues for female patients. This could be reflected in the title: "...survey responses from a largely female sample".

Response:

We agree that this needs to be acknowledged and have placed the number of women who were participants in the abstract and in the limitations bullet point section at the start of the manuscript (page 2-3). We have also provided further contextual information about each quote reported (e.g., age and gender of the participant, whether they were a patient or carer) in the manuscript.

We included a diverse range of patient and carer advisors and liaised with healthcare and community care services in our sampling procedures. We recruited 14 men (16% of the total sample) and two people self-defined as non-binary (2.3%), which is similar to community-based studies of self-harm.³ In the 'Strengths and limitations' subsection of the Discussion section, we have now highlighted the limitation that surveys potentially exclude less represented groups (page 22). The men and non-binary participants who did participate in this study provided rich qualitative descriptions of receiving psychosocial assessments following self-harm in the emergency department. We understand the reviewers concerns but feel it is important not to exclude the voices of these participants. We have kept the title but tried to be as transparent and clear as possible about the comparatively small number of men in the study.

Reviewer comment

10. What was the cultural and linguistic background of participants, and how did this influence the receipt of care in the ED? Our recent paper (see below) in the Australian context, was based around the importance of culture in psychosocial assessments for Aboriginal and Torres Strait Islander people.

Leckning B, Hirvonen T, Armstrong G, Carey TA, Westby M, Ringbauer A, et al. Developing best practice guidelines for the psychosocial assessment of Aboriginal and Torres Strait Islander people presenting to hospital with self-harm and suicidal thoughts. Aust N Z J Psychiatry 2020;48:674-20924082.

Response:

Thank you for highlighting this paper and for this helpful comment. Most participants in our study were White British, which we have highlighted in the 'Strengths and limitations' subsection of the Discussion (page 21-22). Further research is needed to explore how the cultural and linguistic background of participants may influence the receipt of care in the emergency department and in the development of clinical guidelines. This is an important and often neglected area of research. We have written this into the Discussion and cited this paper (page 24).

Reviewer comment

11. Were participant characteristics associated with particular types of responses in any way? This might be hard to reflect on given the lack of diversity in the sample.

Response

The responses given across participant subgroups were generally consistent. We have also provided additional contextual information about each quote reported (e.g., age and gender of the participant, whether they were a patient or carer) in the manuscript, to demonstrate some of this consistency. The responses may be due to restricted diversity or fairly consistent poor experiences when attending the emergency department following self-harm. Other marginalised groups, including those with cultural and linguistic differences, may have had different experiences of care, but more narrowly focussed co-designed studies are needed to investigate this issue in greater detail.

Reviewer comment

12. Were there any discussions around the use of a peer support workforce for some elements of the process?

Response

No, but peer support is widely variable, which is a recent development in liaison psychiatry teams in England.

Reviewer comment

13. The discussion section seems a little light on, and most heavily focused on study limitations rather than discussing the results. For example, one theme that emerged was people's experiences of stigmatising attitudes among clinicians. This is one of a few critically important issues that could have been given their own paragraphs in the discussion section, with relevant references brought in.

Response:

We agree that stigmatising attitudes among clinicians is a critically important issue that has rightly received much attention in the literature. We had a restricted word count and wished to avoid overly duplicating what is reported in other studies but agree this should be highlighted. We have referred to some of this literature and historical context in the Discussion (page 23, under 'Comparisons with existing research').

Reviewer comment

14. Shouldn't the title refer to "psychosocial assessment" rather than "mental assessment". This paper talks about people's experiences being that the assessment was almost exclusively focused on mental health, rather than being a more holistic psychosocial assessment that reflects the issues experienced by the patient. The title unfortunately brings it straight back to mental health again.

Response

Many thanks, this is a very helpful suggestion. We have amended the paper's title accordingly.

Reviewer comment

15. Were there many participants who had no prior history or diagnosis of a mental disorder?

Response

Only four participants had no psychiatric diagnosis. We have now highlighted this in the Discussion and have recommended that further research be conducted in this area (Page 21).

References:

1. Clements C, Turnbull P, Hawton K, Geulayov G, Waters K, Ness J, Townsend E, Khundakar K, Kapur N. Rates of self-harm presenting to general hospitals: a comparison of data from the Multicentre Study of Self-Harm in England and Hospital Episode Statistics. *BMJ Open*. 2016 Feb 1;6(2).
2. Geulayov G, Kapur N, Turnbull P, Clements C, Waters K, Ness J, Townsend E, Hawton K. Epidemiology and trends in non-fatal self-harm in three centres in England, 2000–2012: findings from the Multicentre Study of Self-harm in England. *BMJ Open*. 2016 Apr 1;6(4).
3. Samaritans. Pushed from pillar to post. UK. Samaritans. 2020.

Reviewer: 2

Reviewer Name

Amy Chandler

Institution and Country

University of Edinburgh, UK

Please state any competing interests or state 'None declared':

None declared

Comments to the Author

This is a strong paper that sheds light on an important issue that has long been highlighted as a problem by service users: engagement with staff in A&E when attending for self-harm, and particularly relating to the experience of psychosocial assessment. It is especially encouraging and to be commended that service user/lived experience perspective is so well embedded in project design, analysis and write up. The meaningful involvement of service users/those with lived experience shows – this is a refreshingly written piece that manages to avoid the tendency to objectify and 'other' patients who self-harm that I still see too often in clinically based research of this type.

Response:

Thank you for providing this positive feedback on our manuscript, and for your encouraging comments on our lived experience involvement. We continuously strive and advocate for people having meaningful involvement in research, and so we are particularly grateful to receive this feedback.

Reviewer comment

Using a qualitative survey of this type, with open ended questions, is a suitable method of research to address the research questions. Methods of analysis are well described and suitable. That said, a potential limitation of the online survey approach is that it privileges participants/patients who are more literate and able to express themselves through the written word. This may have inadvertently excluded important perspectives – from patients with more limited grasp of English (whether because English is not their first language, they have language processing difficulties, or they do not like writing). It may be worth further reflecting on how recruitment and study design may further entrench the likelihood of participants being relatively similar (female, white, having a mental health diagnosis). It is good that the authors acknowledge the limitations of their sample being almost entirely composed of white British females. This is indeed a limitation, and also often seen in online studies of self-harm.

Response

We agree with the potential limitation of online surveys. Whilst they provide substantial benefits for inclusion in sensitive research, we agree that caution is required regarding the likely exclusion of people who may be increasingly underrepresented in research. We have now highlighted this issue in the Discussion (Page 22).

We assembled a diverse patient/carer advisory panel in terms of age, gender, socioeconomic position, and ethnicity. We had many meaningful and insightful discussions with the panel during the study's planning and recruitment phases which we enacted during the study. However, recruitment is a known challenge for studies of patients who have harmed themselves and then presented to a hospital emergency department. Part of the issue is funding and capacity. To be truly inclusive, it is likely that we would need to apply several recruitment strategies and conduct multiple separate studies among vulnerable people, including individuals who have a limited understanding of English, those lacking digital knowhow or access, and those who do not like writing or engaging with this type of methodology. One method or study design is rarely enough to capture the range of people who may harm themselves. Future studies need to incorporate several types of research design to include these underrepresented voices, including arts-led designs, but also have funding, time, and capacity to thoroughly recruit participants using different methods and co-designed approaches.

Reviewer:

I note that employment status was measured, but wondered whether this had any impact on responses at all?

Response:

The responses that we received from participants were consistent irrespective of their employment status. This could be a sampling issue or because we asked participants to reflect on their experiences of psychosocial assessment. However, we were unable to probe complex and dynamic issues related to socioeconomic position, class, self-harm, and psychosocial assessment in the survey. We have highlighted this limitation in the Discussion (page 22).

Reviewer comment:

An important contribution of the paper is being able to draw out nuances in how patients experienced psychosocial care. This highlights the vital role of experience, and *how* care is provided. It relates well to existing research which shows relatively enduring negative attitudes towards those who self-harm among clinical practitioners, especially towards some patient groups (e.g. Timson et al 2012). The authors note that 'We are unable to say if substandard or iatrogenically harmful care caused some participants to repeat self-harm'. However, I wanted to push this issue: is it not enough to affirm that receiving care experienced as harmful is harmful, without further 'proving' this via actual further self-harm?

Response

We agree with this important comment and have removed these sentences from the revised manuscript.

Reviewer comment:

Given this issue has been addressed for such a long time, I wonder if a bit more of a historical context could be acknowledged, perhaps through reference to earlier, pioneering user led work on the challenges and harms resulting from poor medical care for people who self-harm e.g. Louise Pembroke, and perhaps also Sam Warner, and Helen Spandler's earlier work here, as well as a study by Hadfield and colleagues that addressed some of the challenges in terms of 'staff attitudes' preventing provision of compassionate care for those who self-harm (Hadfield et al., 2009; Pembroke, 1998; Spandler & Warner, 2007; Warner & Spandler, 2011)?

Response:

We agree with the importance of this pioneering work and are familiar with some of the extensive research on staff attitudes toward people who have harmed themselves. However, we

were constrained by the journal's word count limit. Unfortunately, we had to restrict and cut down the literature considerably to include more of the detailed findings, including lengthy verbatim quotations of whatsome participants had stated in their responses. We have, however, referred to some of this historical literature (e.g., hostile care and patient harm) in the Discussion (page 23-24 under 'Comparisons with existing research').

Reviewer comment

In conclusion the authors call for more resources and training I wonder if this call could be framed in slightly stronger terms, and/or further interrogated: we have had this call for over 30 years... (the first Department of Health guidance for self-harm that I am aware of was published in 1981!). I hope that an article highlighting (again) the harms caused to patients who self-harm when they are treated badly in A&E settings will help, but I must admit to some scepticism that any of this will make a great deal of difference. Earlier reviews by Saunders and colleagues (K. E. Saunders & Smith, 2016; K. E. A. Saunders et al., 2012) found some improvement in staff attitudes following training - but notably patient perspectives were rarely sought. This is a key area that this paper contributes to, along with the focus on experiences of psychosocial assessment specifically. However, I would suggest there is a need to go beyond staff training, and to consider more significant changes, reflecting on how policies, structures and systems, as well as speaking more broadly to societal attitudes towards mental illness might need to shift in order to improve patient experience/care? What about more robust monitoring of patient experience of A&E when attending for mental health/self-harm, as well as active engagement of patients in this process?

Response:

This is an important comment and highlights the broader extent of the issues related to the societal stigma surrounding self-harm and health service training and delivery. The reviewer also highlights a limitation of much health sciences research for self-harm, which is the lack of translation into practice. The implementation gap may be in part due to how we disseminate our research, design, and deliver training, but also the lack of patient involvement and focus on improving patient experience. We have addressed these issues in the Discussion under 'Implications for clinical practice and policy' on page 24.

Reviewer comment

The section where the authors map their recommendations to the NICE guidelines is very helpful, and serves to further reinforce the challenging nature of the study findings, and I imagine may be useful for practitioners or service managers. However, again for me this served to highlight how far there is to go in terms of enacting these guidelines. This also highlights perhaps some challenging aspects of the guidelines themselves, which I acknowledge may be beyond the scope of this particular study. For instance, one of the guidelines addressed is "People who have self-harmed receive the monitoring that they need whilst in the healthcare setting, to reduce self-harm repetition risk." – this focuses attention of healthcare practitioners around surveillance with a view to 'reduction of repetition risk' – which may not sit so neatly with the provision of compassionate care for a patient who might be expected to repeat. Perhaps different measures of treatment success are needed in order to better support and reward practitioners to provide compassionate, non-judgemental care? Overall, this paper makes a strong contribution, and highlights (again) the negative experiences of some patients who self-harm, as well as highlighting how less judgemental and compassionate care can be extremely beneficial. That said, I would suggest the recommendations could be further strengthened - could the authors consider some recommendations for policy as well as practice?

Response:

We agree with these helpful comments. Some of the experiences may reflect poor implementation of current guidelines but we acknowledge that others reflect key challenges not just for guidelines but for clinical practice as a whole – for example, how do we balance the safety of service users with the

provision of care that is as compassionate as it can be? What are the most appropriate outcome measures not just for self-harm research but for clinical services? The NICE guideline for the care treatment of people who have recently harmed themselves are currently under revision, and hopefully these issues will be addressed in the revised version. We are also working closely with services to understand the systems and interpretations of policy guidelines for self-harm. We have now highlighted some possible recommendations for policy in the Discussion (page 24-25, 'Implications for practice and policy').

Reviewer Comment

Future work should also certainly look to move beyond this relatively narrow sample, to engage a more diverse patient group.

Response:

We agree and have highlighted this issue in the Discussion and also in the aforementioned comments.

VERSION 2 – REVIEW

REVIEWER	Dr Gregory Armstrong University of Melbourne, Australia
REVIEW RETURNED	01-Feb-2021

GENERAL COMMENTS	The authors have nicely addressed the issues raised during peer review. I recommend the manuscript is accepted for publication.
---

REVIEWER	Amy Chandler University of Edinburgh, UK
REVIEW RETURNED	16-Feb-2021

GENERAL COMMENTS	Many thanks to the authors for their careful and considered responses to the points raised in the review. I am satisfied that these have been addressed more than adequately. The paper makes an important contribution to knowledge, with clear and helpful recommendations for policy and practice. One minor point to address prior to publication - would suggest editing the final bullet point in the article summary for clarity - suggested edit here: "• Most of the respondents in the study were White British Women from England (72/88, 81.8%) and their experiences may differ in important ways from other patients who have limited literacy skills, or who did not complete the survey.
---

VERSION 2 – AUTHOR RESPONSE

Reviewer: 1

Dr. Gregory Armstrong, University of Melbourne

Comments to the Author:

The authors have nicely addressed the issues raised during peer review. I recommend the manuscript is accepted for publication.

Response

Many thanks for your valuable comments and suggestions during the peer review process.

Reviewer: 2

Dr. Amy Chandler, The University of Edinburgh

Comments to the Author:

Many thanks to the authors for their careful and considered responses to the points raised in the review. I am satisfied that these have been addressed more than adequately. The paper makes an important contribution to knowledge, with clear and helpful recommendations for policy and practice.

One minor point to address prior to publication - would suggest editing the final bullet point in the article summary for clarity - suggested edit here:

"• Most of the respondents in the study were White British Women from England (72/88, 81.8%) and their experiences may differ in important ways from other patients who have limited literacy skills, or who did not complete the survey.

Response:

Thanks for your valuable comments and suggestions to improve the manuscript throughout the peer review process. We have now amended this sentence in the revised manuscript.